# Stellate Ganglion Block for PTSD and Chronic Low Back Pain: A Case Report of Three Veterans

**DOI:** 10.3390/jcm14103375

**Published:** 2025-05-12

**Authors:** Lindsay Sterling, Kristy Fisher, Anna Woodbury

**Affiliations:** 1Department of Anesthesiology, Division of Pain Medicine, Emory University, Atlanta, GA 30322, USA; lindsay.erin.sterling@emory.edu (L.S.); krstyfshr@gmail.com (K.F.); 2Division of Pain Medicine, Atlanta VA Healthcare System, Decatur, GA 30033, USA

**Keywords:** post-traumatic stress disorder, chronic low back pain, central sensitization

## Abstract

**Background**: Stellate Ganglion Blocks (SGB) involve injecting local anesthetic near the stellate ganglion, which includes the C6, C7, and T1 ganglia. This procedure induces a sympathetic blockade and has been employed to address various conditions, such as post-traumatic stress disorder (PTSD), ventricular arrhythmias, and chronic pain syndromes like complex regional pain syndrome (CRPS). Central to this case series is the exploration of SGB as a unified treatment for PTSD and chronic low back pain—two conditions linked by central sensitization. **Case Series Overview**: The study presents three female veterans with histories of PTSD, chronic low back pain, and myofascial pain. These patients had not responded to conventional treatments, including medications and interventional procedures. They underwent SGB with a combination of 10 mg preservative-free dexamethasone sodium phosphate, 4 mL preservative-free lidocaine 2% with epinephrine, and 1 mL preservative-free bupivacaine 0.25%. The procedure was well tolerated without adverse effects. All three patients experienced significant improvements. The first and third patients reported reductions in PTSD symptoms, low back pain, and myofascial pain. The second patient experienced relief from PTSD symptoms and prolonged reduction in myofascial pain. This case series is the first to document SGB’s effectiveness in treating chronic low back pain alongside PTSD. **Conclusions**: The findings suggest that SGB could be an effective therapy for chronic overlapping conditions like PTSD, chronic low back pain, and myofascial pain, all of which share central sensitization mechanisms. The literature supports the notion that these conditions involve both physical and psychiatric components potentially responsive to SGB. By targeting sympathetic hyperactivity and reducing norepinephrine levels, SGB may alleviate symptoms across these interconnected syndromes. This case series highlights the potential of SGB as a novel approach for managing comorbid PTSD and chronic pain conditions. Further research is warranted to confirm its efficacy and explore its broader applications in treating central sensitization-related disorders and chronic overlapping pain conditions (COPC), potentially responsive to sympathetic blockade.

## 1. Introduction

Stellate ganglion blocks (SGB) have effectively been utilized in both treatment resistant post-traumatic stress disorder (PTSD) [1], as well as with refractory chronic pain in the upper extremities, head, and neck [2]. In the field of chronic pain management, SGB has been most utilized in the treatment of complex regional pain syndrome (CRPS) [3]. The overlapping pathophysiology of PTSD and chronic pain syndromes such as low back pain, CRPS, and chronic myofascial pain may contribute to the success of the SGB in alleviating two seemingly distinct indications.

PTSD and chronic pain syndromes, like chronic low back pain (cLBP), share mechanisms involved in central sensitization (CS). This partially explains the high comorbidity rate, particularly in veterans. A 2023 study of over 5 million VA service users found that over half of those diagnosed with PTSD also met the criteria for chronic pain [4]. One mechanism is the Shared Vulnerability model, where anxiety sensitivity is a predisposing factor. Anxiety sensitivity increases fear and avoidance of pain-inducing activities, raising the risk of chronic pain [5]. The mutual maintenance model also identifies factors that maintain both chronic pain and PTSD, including attentional biases, avoidance coping, and pain as a trauma reminder; the cognitive demands from the symptoms of pain and PTSD may limit the use of healthy coping strategies [5].

Anatomical studies reveal gray matter atrophy in brain areas related to stress, pain response, and memory (medial prefrontal cortex, hippocampus, and insular cortex) in patients with PTSD and chronic pain [6]. The amygdala, which processes threat and fear, is hypertrophied and hyperactive in both conditions. Immune system dysregulation is also linked, with increased pro-inflammatory cytokines and cortisol receptor reactivity in PTSD patients and those with chronic pain.

CS involves heightened response in nociceptive pathways due to increased excitability and reduced inhibitory modulation [7] and can manifest clinically in patients as hyperalgesia and allodynia. While CS has been proven in animal models through electrophysiological recordings [8], studies have found a higher degree of acute central sensitization in individuals with PTSD [7], as well as cLBP [9]. cLBP and PTSD are both categorized as central sensitization syndromes (CSS) (Figure 1), along with disorders such as fibromyalgia, myofascial pain syndrome, and CRPS [10].

SGB can influence CS through multiple mechanisms, making it a potential treatment for comorbid PTSD and chronic pain syndromes. SGB inhibits sympathetic nervous system activity, decreasing central nervous system hyperexcitability seen in CS [11]. This blockade may disrupt abnormal connections between the sympathetic and sensory systems that can develop after nerve injury [12], potentially helping to reduce the afferent nociceptive stimuli contributing to CS. SGB can also reduce sympathetic activity by reducing nerve growth factor (NGF), associated with acute and chronic stress response. Decreased NGF leads to decreased norepinephrine levels, mitigating peripheral nociceptor sensitization [11]. SGB also vasodilates innervated areas [12], improving circulation, which may reduce inflammation and promote healing. The stellate ganglion has extensive neuronal connections to brain regions involved in pain processing, such as the hypothalamus and amygdala [12], and may influence the neuroplastic changes associated with CS through modulation of these connections. SGB also has anti-inflammatory properties, which may help reduce neuroinflammation, a key component of CS.

This is the first case series to demonstrate stellate ganglion block (SGB) as a treatment for both PTSD and cLBP simultaneously.

## 2. Case Series

Patients were assessed for PTSD symptoms by a psychiatrist using the Clinician-Administered PTSD scale for DSM-5 (CAPS-5) and at treatment visits and via telephone follow-up. CAPS-5 is a structured interview that makes a current diagnosis of PTSD, as well as assesses PTSD symptoms over the past week. Standardized questions are used to target the duration and onset of symptoms, impact of symptoms on daily functioning, and change in symptoms. For each symptom, severity is rated on a five-point scale: 0 (absent), 1 (mild/subthreshold), 2 (moderate/threshold), 3 (severe/markedly elevated), and 4 (extreme/incapacitating). Pain levels were similarly assessed at visits and over the phone. Patients were asked to rate their pain on a numerical pain rating scale (NPRS) from 0 to 10, with 0 being no pain and 10 being the worst pain possible for each type of pain they were experiencing. CS etiologies were assessed for each patient (Figure 1) and SGB was performed as described in the legend of Figure 2.

**Patient 1**: A 40-year-old African American female veteran with treatment-resistant PTSD, migraines, and chronic myofascial pain presented with low back pain and bilateral, non-radiating neck and posterior arm pain. Imaging revealed multilevel degenerative discogenic disease and neural foramina narrowing. Despite extensive treatment—including duloxetine, gabapentin, epidural steroid injections, radiofrequency ablation, and peripheral nerve stimulation—her pain persisted. PTSD management with SSRIs, prazosin, topiramate, and gabapentin also failed to provide relief. The patient had participated in psychotherapy and trauma-focused recovery programs. Given the refractory nature of her symptoms, a unilateral stellate ganglion block (SGB) was recommended to address both PTSD and upper extremity pain (Figure 2). On the day of the procedure, her pain was rated 8/10 for both upper extremity and lower back pain, while PTSD symptoms were classified as “severe” on the CAPS-5 scale. Day 1 post procedure, her pain dropped to 0/10 in both areas, and PTSD symptoms became undetectable (Table 1). One week later, her upper extremity pain increased to 5/10 and lower back pain to 4/10, but PTSD symptoms remained absent. At six weeks post-procedure, she experienced a slight recurrence of PTSD symptoms and low back pain, though both were improved compared to baseline. By eight weeks, her low back pain returned, prompting continued treatment with bilateral L5-S3 radiofrequency ablations. Remarkably, one year after the initial SGB, she reported sustained reduction in PTSD symptoms with no recurrence of upper extremity pain.

**Patient 2**: A 55-year-old Caucasian female veteran with history of PTSD, bipolar disorder, and conversion disorder presented with over 20 years of chronic low back pain. She underwent a L3-S1 Laminectomy in 2008, which resolved her pain for about 3–4 years before symptoms of low back and leg pain returned. The patient had failed trials of multiple NSAIDs, and at time of procedure, her medication regimen included duloxetine 120 mg daily, gabapentin 800 mg three times daily, methocarbamol 2–3 times a week at night, ibuprofen 800 mg 4–5 times daily, and topiramate 25 mg daily for headaches. The patient had received multiple epidural steroid injections but reported irritability and angry mood swings after injections. In addition to her low back pain, she experienced symptoms of PTSD and left shoulder myofascial pain. The decision was made to undergo a unilateral SGB for pain and PTSD symptoms. Her pre-procedure pain scores were 8/10, and post-procedures scores were 6/10 with improved range of motion of her left arm. She was unable to be contacted for her 1- and 6-week follow-ups. At 3 months, she reported improvement in her PTSD and shoulder symptoms, but no improvement in her low back pain. She elected to repeat the SGB 7 months after her initial procedure for continued shoulder pain and PTSD relief.

**Patient 3**: A 29-year-old Caucasian female veteran with severe PTSD presented to the pain clinic with chronic low back pain, myofascial shoulder pain, and chronic headaches. Despite multiple treatments, including physical therapy and various medications, her symptoms persisted. Previous interventions like sacroiliac joint injections and caudal epidural steroid injections provided limited or no relief. A Stellate Ganglion Block (SGB) was recommended to address her central sensitization syndromes and COPC. Pre-procedure, her pain scores were 10/10 for back pain and 6/10 for headache and shoulder pain, with PTSD symptoms in the “severe” range on CAPS-5. Immediately post-SGB, her headache pain reduced to 1/10 and shoulder pain to 0/10, while back pain and PTSD symptoms remained unchanged. At one-week follow-up, she reported significant improvement in PTSD symptoms, including decreased sensitivity to triggers. Pain scores for low back, headache, and shoulder were all 0/10. At 12 weeks, she received a repeat SGB with similar positive results. Subsequently, she underwent Bilateral L5-S3 Sacral Radiofrequency Ablation for sustained low back pain relief, which she tolerated better post-SGB. Four months later, when her low back pain returned to 10/10, the patient opted for another SGB over sacral RFA, prioritizing PTSD symptom relief with the added benefit of pain reduction. She has since been receiving these injections every 3–4 months, with similar benefit.

## 3. Discussion

### 3.1. Anatomy of the Stellate and Applications of Its Blockade

The Stellate Ganglion (SG) is a sympathetic ganglion situated over the 6th (C6) and 7th (C7) cervical vertebrae and first thoracic vertebrae (T1); its exact position can vary slightly among individuals. The SG plays a crucial role in sympathetic innervation of the head, neck, upper extremities, and upper thorax, and serves as a relay station for sympathetic nerve fibers, influencing various regional physiological functions, including blood flow, sweating, and pain perception. SGB has been used in clinical practice since the 1940s and involves injecting local anesthetic near the sympathetic ganglia for a broad sympathetic blockade [13]. SGB has been used for a wide variety of seemingly unrelated pathologies, including complex regional pain syndrome (CRPS), PTSD, ventricular tachyarrhythmias, and to create an extremity blockade and vasodilation for surgical procedures requiring regional anesthesia. Advances in ultrasound technology and fluoroscopy have made SGB safer and more effective, leading clinicians to extrapolate its sympathetic blocking properties to other conditions [11].

The underlying mechanisms of PTSD and conditions such as CRPS may seem unrelated, but their pathophysiologies can be linked under CS (Figure 1), a term covering a range of clinical disorders related to the amplification of pain reception by the central nervous system, which can occur independently of direct result or injury [14]. SGB has been widely used, particularly in military settings, for PTSD due to its ability to blunt the increased noradrenergic activity of PTSD. In a multi-site trial of over 100 service members suffering from PTSD, 2 SGB treatments 2 weeks apart were effective in reducing PTSD severity at a clinical threshold over 8 weeks [1]. Similarly, conditions such as CRPS, which involve a heightened pain response to a physical insult, have been treated with SGBs due to its ability to blunt sympathetic crosstalk with nociceptors [13]. There are currently no case reports that involve the treatment of both PTSD and chronic pain in the same patient. One study proposed a mechanism for the underlying efficacy for SGBs in CRPS and PTSD, which involved the increased release of NGF in pathological states, leading to an increase in Norepinephrine and sympathetic hyperactivity. The study suggested that the efficacy of the SGB lies in its ability to block this cascade [15].

### 3.2. Safety of the Stellate Ganglion Block

SGB is generally considered a safe procedure, especially when performed under imaging guidance, but it does carry some risks. There were no reported unanticipated adverse effects in this case series. The incidence of severe complications in found in a 1992 study was approximately 1.7 per 1000 blocks [16]. This research was performed prior to the advent of fluoroscopic and ultrasound visualization, which have greatly increased the safety of the procedure. Common side effects include hoarseness, lightheadedness, and temporary Horner’s syndrome symptoms such as drooping eyelids and bloodshot eyes [17]. More serious but rare complications can include infection, bleeding, nerve damage, seizures, and in extremely rare cases, pneumothorax or allergic reactions [18]. Vascular disruption during the procedure can lead to transient locked-in syndrome, hemorrhage, or hematoma formation [19]. The risk of these complications is significantly reduced when SGB is performed by experienced practitioners using ultrasound or fluoroscopic guidance [17]. While one study reported a case of death due to massive hematoma leading to airway obstruction, such severe outcomes are exceedingly rare [20]. Overall, when performed by qualified professionals with proper imaging guidance, SGB remains a low-risk procedure with potential benefits for various conditions.

### 3.3. Strengths

The most strongly validated treatments for PTSD are trauma-focused psychotherapies—including prolonged exposure (PE), cognitive processing therapy (CPT), and eye movement desensitization and reprocessing (EMDR)—with selective serotonin reuptake inhibitors (SSRIs) such as sertraline and paroxetine also conditionally recommended as pharmacologic options. While SGB is not yet, in general, the clinical guideline for PTSD, it has been adopted into clinical practice and Veterans Affairs guidance for refractory PTSD based on recent literature, including a sham-controlled study of over 100 active-duty service members randomized to SGB or sham in a 2:1 ratio [1].

While SGBs have been used for PTSD, the true novelty of this case series lies in the improvement of two of the patients’ myofascial pain, including cLBP. Chronic low back pain is one of the most common complaints in the field of chronic pain, and in the absence of anatomical pathology can be difficult to treat. Significant studies have shown the presence of CS in patients with cLBP [9]. Pressure algometry studies have shown patients suffering from cLBP have significantly lowered pressure pain thresholds and hyperalgesia both in their back and peripheral sites, and individuals with cLBP have decreased pain thresholds, increased pain responses, and prolonged duration of pain after heat, chemical, pressure, and electrical stimuli, all consistent with features of CS [9]. Despite the inclusion of cLBP of mechanical origins as a CS syndrome, the SGB has not previously been studied for treatment of cLBP or myofascial pain syndrome, and thus we present novel results for further exploration and research.

This case series also presents a novel approach to addressing multiple conditions—cLBP, myofascial pain, and PTSD—with a single intervention. Prior reports have utilized SGB for simultaneous treatment of PTSD and reflex sympathetic dystrophy (now known as complex regional pain syndrome), but not for more distal pain [21]. In this case series, all three patients had multiple conditions indicating CS (Figure 1), nociplastic pain, and hyperactive sympathetic activity that improved after SGB was performed. All three patients had a reduction in their PTSD and myofascial pain symptoms, while two additionally had a reduction in their cLBP severity.

### 3.4. Limitations

Our findings are limited by a lack of generalizability due to the small sample size and absence of a control group, making it difficult to apply the results to larger populations. Additionally, case series cannot establish causal relationships between interventions and outcomes, leaving room for confounding factors to influence the observed effects. The absence of a comparison group further complicates efforts to determine whether observed outcomes are attributable to the intervention or other factors. While it is unlikely a placebo response, given these patients’ non-response to multiple other modalities including interventional procedures, it must be acknowledged that there was no placebo control group. These inherent limitations underscore the need for cautious interpretation of our findings and highlight the importance of further research using more robust study designs.

This case series demonstrates simultaneous improvement in PTSD and chronic pain following SGB, but personality-related vulnerabilities may explain divergent treatment responses observed in broader clinical contexts. While SGB targets shared neurobiological mechanisms (e.g., sympathetic hyperactivity and central sensitization), the persistence of certain psychological traits could lead to asymmetrical outcomes where one condition improves without the other; for example, cognitive fusion with trauma memories might diminish through SGB’s neurobiological effects, while unaddressed pain-related thought patterns (e.g., “My back is irreparably damaged”) sustain disability through psychological inflexibility.

### 3.5. Future Directions

Future research should aim to identify the optimal injectate solution for SGBs. This case series used a 6 mL solution of dexamethasone, lidocaine 2% with epinephrine, and bupivacaine 0.25%, which provided varying durations of pain relief among patients. However, limited data exist on the ideal composition, volume, and concentration of injectates to maximize the duration of relief. Challenges in maintaining consistent patient follow-up in this study further limited the accuracy of pain relief duration data. Additional research is needed to clarify how different injectate formulations influence outcomes. Another key area for investigation is determining which patients are most likely to benefit from SGBs; cLBP has diverse causes, often requiring targeted treatments based on specific anatomical or pathological factors. The effectiveness of SGB compared to or in combination with traditional interventions should be explored, particularly in cases involving CS. While this case series focused on patients with both cLBP, myofascial pain, and PTSD, future studies should evaluate whether SGB is more effective with refined patient selection and optimized treatment protocols. Mulvaney et al. have provided some guidance regarding this topic [22].

Further, the concurrent symptom relief observed her may reflect baseline resilience factors (patients potentially had lower premorbid neuroticism scores than non-responders in other studies), though this was not quantified in the present investigation. In the future, we would recommend administering personality assessments pre/post-SGB to quantify trait modulation, comparing SGB outcomes in patients with/without borderline personality disorder traits, given their association with somatic symptom persistence, and investigating whether personality vulnerabilities moderate the duration of SGB’s effects. These observations align with the triple vulnerability model, suggesting that while SGB addresses biological and generalized psychological vulnerabilities through autonomic modulation, residual disorder-specific cognitive patterns may require targeted cognitive-behavioral approaches for sustained remission.

### 3.6. Conclusions

This case series is the first to demonstrate the successful treatment of both PTSD and cLBP/myofascial pain in individual patients using a single intervention. The results reinforce the connection between the pathophysiology of PTSD and chronic pain conditions and suggest the use of SGB as a unified treatment approach for these cooccurring conditions. While further research needs to be performed, these results suggest that SGB could be a singular treatment option for patients suffering from both PTSD and chronic pain, potentially simplifying treatment regimens for complex cases involving CS syndromes.

## Figures and Tables

**Figure 1 jcm-14-03375-f001:**
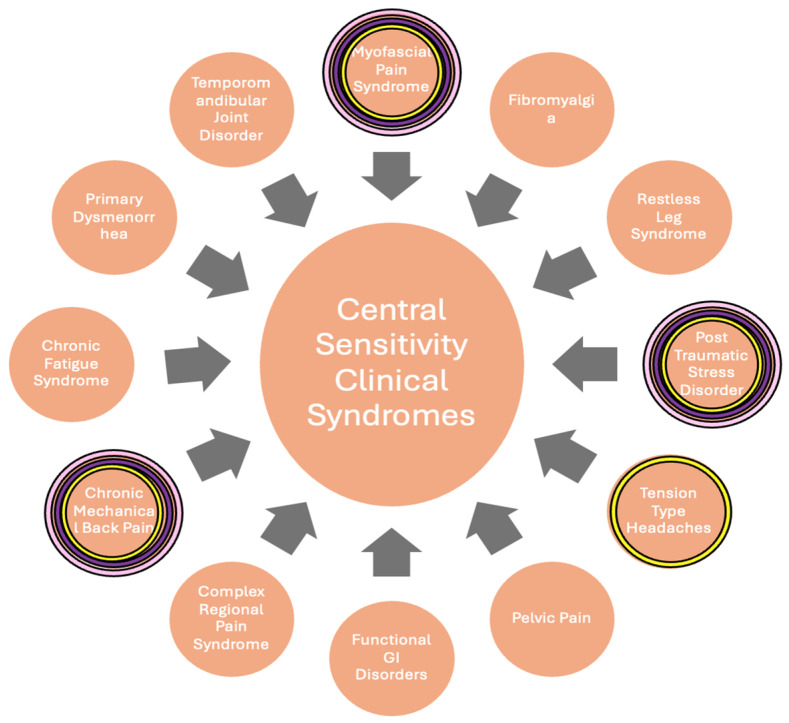
**Central sensitization syndromes**. The image denotes overlapping CSS, and colored highlighting reflect the syndromes present in this case series, as follows: Pink: Patient 1. Purple: Patient 2. Yellow: Patient 3. Each patient had more than one syndrome.

**Figure 2 jcm-14-03375-f002:**
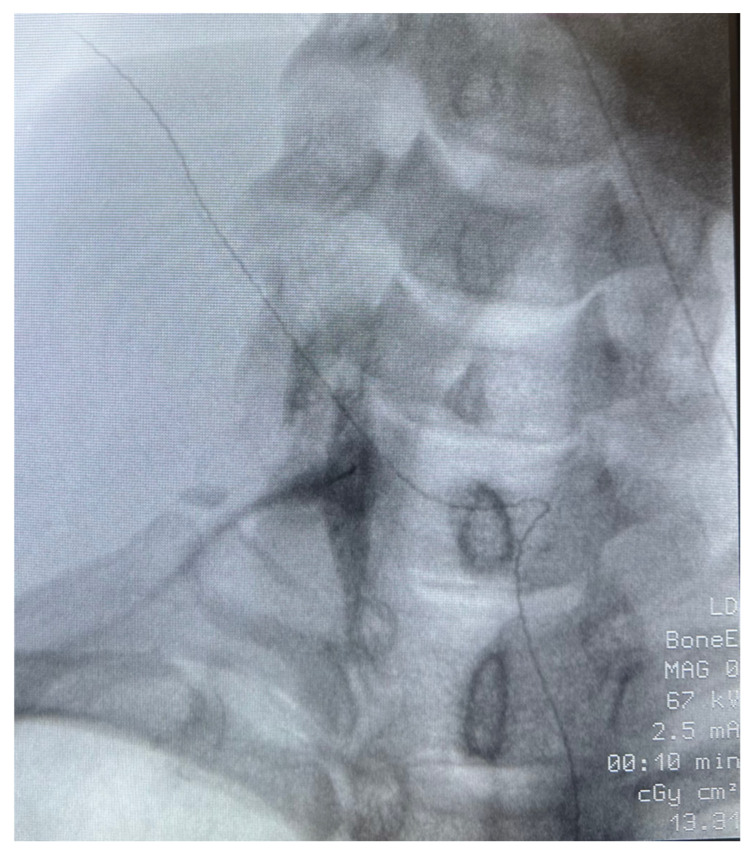
**Fluoroscopically guided right stellate ganglion block** (Patient #1). The same technique was used for Patient 2 and 3. Patients were placed supine with neck in an extended position. The neck and shoulder area were prepped and draped in a sterile fashion. Using anteroposterior fluoroscopy, an image of the patient’s neck was obtained. The C6 transverse process (Chassaignac’s tubercle) was identified and marked. The skin over the area was topicalized with 1% lidocaine. The carotid artery and trachea were identified and palpated. Lateral traction on the carotid was placed, and a 25 gauge, 3.5 inch spinal needle was inserted and advanced, guided by fluoroscopy, to the target of the vertebral body and the uncinate process of C6. Once the target was reached, and after negative aspiration for heme, air, and cerebrospinal fluid, and in the absence of paresthesias, 1 mL of Omnipaque 180 mg/mL was injected to confirm spread along the stellate ganglion. A second aspiration was performed. When confirmed negative, 6 mL total of a mixture of dexamethasone 10 mg, lidocaine 2% with epinephrine, and bupivacaine 0.25% with epinephrine, was injected in 2–3 mL increments at the stellate ganglion, with frequent monitoring of signs and symptoms of intravascular injection.

**Table 1 jcm-14-03375-t001:** **Longitudinal changes in pain and PTSD.** Pain scores using the numerical pain rating score and PTSD symptomatology scored using CAPS-5 for all three patients preprocedure: one day post procedure, one week post procedure, and six weeks post procedure.

	Pre-Procedure		1 Day Post Procedure		1 Week Post Procedure		6 Weeks Post Procedure	
	Pain	PTSD	Pain	PTSD	Pain	PTSD	Pain	PTSD
**Patient 1**	Neck/Arm: 8	Severe/Markedly Elevated	Neck/Arm: 0	Mild/Sub-Threshold	Neck/Arm: 5	Mild/Sub-Threshold	Neck/Arm: 5	Moderate/Threshold
	Back: 8		Back: 0		Back: 4		Back: 6	
**Patient 2**	Back: 8	Severe/Markedly Elevated	Back: 6	Severe/Markedly Elevated	Not Available			
	Shoulder: 8		Shoulder: 6					
**Patient 3**	Back: 10	Severe/Markedly Elevated	Back: 6	Severe/Markedly Elevated	Back: 0	Severe/Markedly Elevated	Not Available	
	Upper Extremity: 6		Upper Extremity: 0		Upper Extremity: 0			
	Shoulder: 6		Shoulder: 0		Shoulder: 0			

## Data Availability

The original contributions presented in this study are included in the article. Further inquiries can be directed to the corresponding author.

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
