# Peer review of "Stellate Ganglion Block for PTSD and Chronic Low Back Pain: A Case Report of Three Veterans"

_jcm, 2025, doi:10.3390/jcm14103375_

Round 1

Reviewer 1 Report

Comments and Suggestions for Authors

The case series is novel. 

Line 14- myofascial pain indicative of nociplastic pain? Did all three patients have this?

What are practical difference between the shared vulnerability and mutual maintenance models?

On line 66 you start writing what SGB can do for CS disorders. Please review the placebo controlled studies for SGB for PTSD and SGB for nociplastic pain. SGB is not on the guidelines for PTSD. Is it on any guidelines for nociplastic pain (I see in the discussion you write it is not researched for chronic low back pain)?

Patient 1. What psychotherapy treatments did she fail prior to the SGB? Was the SGB unilateral or bilateral?

Patient 2. Was the SGB unilateral or bilateral. Was she on any medications? Did she have any other psychiatric diagnoses other than PTSD. 

Line 180 there is a 9

Figure 1 legend- what is the distinction between disease and illness? What are the different colors of the rings indicating?

Readers find it interesting if you would expand on overlapping vulnerability characteristics from a personality perspective (not just central sensitization) between the two disorders (chronic pain and PTSD). Especially if the temporal dynamics of response don't track (PTSD better but not pain or vice versa). Maybe it is a placebo response....

Author Response

We thank the reviewers for their constructive comments on the manuscript and appreciate recognition of the many strengths in its originality, as well as recommendations for improvement. We have addressed the reviewers’ concerns, below, and in red in the text.

Reviewer 1

The case series is novel. 

Comment 1: Line 14- myofascial pain indicative of nociplastic pain? Did all three patients have this?

Response 1: This has been corrected and expanded throughout the manuscript; yes all 3 had a component of myofascial pain

Comment 2: What are practical difference between the shared vulnerability and mutual maintenance models?

Response 2:

The shared vulnerability model and mutual maintenance model differ in their focus and mechanisms for explaining the comorbidity of PTSD and chronic pain. Shared vulnerability: Emphasizes pre-existing risk factors (e.g., anxiety sensitivity, hyperarousal tendencies, somatization) that predispose individuals to both conditions simultaneously. Mutual maintenance: Focuses on bidirectional reinforcement between PTSD and pain symptoms after onset (e.g., pain triggering trauma memories, PTSD symptoms exacerbating pain perception).

Comment 3:  On line 66 you start writing what SGB can do for CS disorders. Please review the placebo controlled studies for SGB for PTSD and SGB for nociplastic pain. SGB is not on the guidelines for PTSD. Is it on any guidelines for nociplastic pain (I see in the discussion you write it is not researched for chronic low back pain)? 

Response 3: It is not in general clinical guidelines as of yet, however it is being adopted into clinical practice for refractory PTSD based on recent literature, specifically the large RCT in active duty military members that has been cited in this report (Olmsted, JAMA Psychiatry, 2020). Here is a link to VA guidance regarding SGB for PTSD: chrome-extension://efaidnbmnnnibpcajpcglclefindmkaj/https://www.va.gov/HEALTHPARTNERSHIPS/resources/SGBforPTSD_508.pdf. We have expanded this discussion in section 3.3.

Comment 4: Patient 1. What psychotherapy treatments did she fail prior to the SGB? Was the SGB unilateral or bilateral?

Response 4: Unilateral; clarified in page 2, line 97. Psychotherapy mentioned on line 95-96.

Comment 5: LS/AW Patient 2. Was the SGB unilateral or bilateral. Was she on any medications? Did she have any other psychiatric diagnoses other than PTSD. 

Response 5: Unilateral mentioned page 3, line 117. Her medication regimen is listed in page 3, lines 112-114. Bipolar disorder, and conversion disorder added to PTSD diagnosis.

Comment 6: Line 180 there is a 9 –

Response 6: It’s a reference number, added the brackets

Comment 7: Figure 1 legend- what is the distinction between disease and illness? What are the different colors of the rings indicating?

Response 7: The legend has been fixed and figure changed.

Comment 8: Readers find it interesting if you would expand on overlapping vulnerability characteristics from a personality perspective (not just central sensitization) between the two disorders (chronic pain and PTSD). Especially if the temporal dynamics of response don't track (PTSD better but not pain or vice versa). Maybe it is a placebo response....

Response 8: Placebo acknowledgement has been added in limitations, as well as a second paragraph with a discussion of personality-related vulnerabilities. This has also been expanded upon with a second paragraph in “Future Directions,” though this is now well over the word limit.

Reviewer 2 Report

Comments and Suggestions for Authors

Strengths and General Comments

The main strength of the article is certainly the originality of the topic, which has been developed with a good synthesis of the existing literature, supported by solid neurobiological models such as shared vulnerability and mutual maintenance. The manuscript provides a thorough description of the anatomy, mechanisms of action, and safety of SGB and the references are up-to-date and appropriate. The English language is correct and easy to understand. The article would benefit from the inclusion of a dedicated section that thoroughly explains the materials and methods used in the study. This would provide a clear framework for replicating the cases presented, ensuring transparency and reproducibility. Additionally, the article would gain value from a concise conclusionthat explicitly responds to the objectives outlined by the authors, summarizing the key findings and linking them back to the study's initial goals.

ABSTRACT

  • The study objectives are not clear. It is stated that the aim of the study is to evaluate the efficacy of SGB in PTSD and low back pain (lines 12–13, 80–81). However, in lines 25–31, 155–156, 193–194 and 187 , a more general outcome is reported. This broader objective would make the presentation of the three cases and the discussion more coherent, as the effects are measured across various central pain conditions. Actually in 1 case of 3 (Case 2) the improvement concerns shoulder pain and PTSD, with no improvement in the low back pain component (lines 112–113).
  • Lines 14–25: A brief overview of currently validated treatments for PTSD would be useful.

INTRODUCTION

  • The introduction is clearly written, and the literature review is well-structured, providing a logical and coherent assumption for the formulated hypothesis.

CASE REPORT

  • There is no explanation of the methodology or the technique used to perform the block (fluoroscopy? ultrasound?). The follow-up timings are inconsistent across the three cases. The "Case Report" section would benefit from a more structured format applied to all three cases (e.g. diagnosis, previous treatments, results in terms of response to SGB, and follow-up at 1 day, 1 week and 6 weeks), in order to enhance methodological rigor.
  • Line 93: The CAPS-5 is mentioned, but there is no explanation of what it is or how it was used systematically during follow-up. Similarly, there is no description of the method used to assess pain or how it was consistently applied during follow-ups.
  • Information on the reproducibility of the technique is lacking (e.g., after how much time can it be repeated? Did the described experience help to suggest a treatment protocol?).

Minor Comments

  • The citation of Figure 2 could benefit from an explanation of the technique used to reach the ganglion (e.g. fluoroscopy). Was the same technique used for the other two cases?
  • Line 93: It is unclear when the measurement was taken, on the same day or the following day (see Table 1).
  • Table 1: What does "RUE" mean?

Safety of the Stellate Ganglion Block

  • Adverse effects following the procedure in the reported cases are not discussed.

CONCLUSION

  • Lines 194-196: The data concern only three patients (all female veterans), they are variable, and include incomplete follow-ups. Therefore, the findings are not generalizable and should be considered as preliminary, serving to prompt further research with more robust evidence.

Author Response

Reviewer 2

Comment 1: The main strength of the article is certainly the originality of the topic, which has been developed with a good synthesis of the existing literature, supported by solid neurobiological models such as shared vulnerability and mutual maintenance. The manuscript provides a thorough description of the anatomy, mechanisms of action, and safety of SGB and the references are up-to-date and appropriate. The English language is correct and easy to understand. The article would benefit from the inclusion of a dedicated section that thoroughly explains the materials and methods used in the study. This would provide a clear framework for replicating the cases presented, ensuring transparency and reproducibility. Additionally, the article would gain value from a concise conclusion that explicitly responds to the objectives outlined by the authors, summarizing the key findings and linking them back to the study's initial goals.

Response 1: Due to word count limitations, a Methods section was not added for this case series. However, the procedure methods are expanded upon in the legend for Figure 2.

A conclusion section was added.

ABSTRACT

Comment 2: The study objectives are not clear. It is stated that the aim of the study is to evaluate the efficacy of SGB in PTSD and low back pain (lines 12–13, 80–81). However, in lines 25–31, 155–156, 193–194 and 187 , a more general outcome is reported. This broader objective would make the presentation of the three cases and the discussion more coherent, as the effects are measured across various central pain conditions. Actually in 1 case of 3 (Case 2) the improvement concerns shoulder pain and PTSD, with no improvement in the low back pain component (lines 112–113).

Response 2: We have expanded the discussion of cLBP to include a more general discussion on myofascial pain.

Comment 3: Lines 14–25: A brief overview of currently validated treatments for PTSD would be useful.

Response 3: This has been added at the beginning of section 3.3

INTRODUCTION

  • The introduction is clearly written, and the literature review is well-structured, providing a logical and coherent assumption for the formulated hypothesis.

CASE REPORT

Comment 4: There is no explanation of the methodology or the technique used to perform the block (fluoroscopy? ultrasound?). The follow-up timings are inconsistent across the three cases. The "Case Report" section would benefit from a more structured format applied to all three cases (e.g. diagnosis, previous treatments, results in terms of response to SGB, and follow-up at 1 day, 1 week and 6 weeks), in order to enhance methodological rigor.

Response 4: This has been expanded upon in the legend for Figure 2. A brief methodology description has been added at the beginning of section 2. As for the inconsistent follow ups, that was due to inconsistent patient response and was discussed as a limitation of the study in page 5 line 206.

Comment 5: Line 93: The CAPS-5 is mentioned, but there is no explanation of what it is or how it was used systematically during follow-up. Similarly, there is no description of the method used to assess pain or how it was consistently applied during follow-ups.

Response 5: The CAPS-5 is a structured interview that makes a current diagnosis of PTSD, as well as assesses PTSD symptoms over the past week. Standardized questions are used to target the duration and onset of symptoms, impact of symptoms on daily functioning, and change in symptoms. We have included this as an asterisk under Table 1. Because all authors are clinicians, a clinical assessment tool was used. This was a case series, not a prospective trial.

Comment 6: Information on the reproducibility of the technique is lacking (e.g., after how much time can it be repeated? Did the described experience help to suggest a treatment protocol?).

Response 6: Every 3 months, on average due to our clinical load; now reported for Patient 3, page 3, line 140.  Prior literature describes the effects lasting for at least 6 weeks (Olmsted et al.).

Minor Comments

Comment 7: The citation of Figure 2 could benefit from an explanation of the technique used to reach the ganglion (e.g. fluoroscopy). Was the same technique used for the other two cases?

Response 7: Thank you, this has been fixed.

Comment 8: Line 93: It is unclear when the measurement was taken, on the same day or the following day (see Table 1).

Response 8: Thank you, this has been clarified in Table 1.

Comment 9: Table 1: What does "RUE" mean?

Response 9: This was a shorthand abbreviation for “right upper extremity” we use in clinical notes, but has now been adjusted to spell out “upper extremity.”

Safety of the Stellate Ganglion Block

Comment 10: Adverse effects following the procedure in the reported cases are not discussed.

Response 10: No unanticipated adverse effects were reported; this can now be found on page 4 line 173-174

CONCLUSION

Comment 11: Lines 194-196: The data concern only three patients (all female veterans), they are variable, and include incomplete follow-ups. Therefore, the findings are not generalizable and should be considered as preliminary, serving to prompt further research with more robust evidence. 

Response 11: Agree, and further highlighted in the limitations section on page 5, lines 221-223.

Reviewer 3 Report

Comments and Suggestions for Authors

This manuscript illustrates findings among 3 individuals with concurrent posttraumatic stress disorder (PTSD) and central sensitization (CS) who were treated with stellate ganglion block (SGB). It presents a rationale for how these conditions relate to one another, and as such, how SGB might be appropriate for both. This is an intriguing subject area for consideration.

Strengths. The description of the Shared Vulnerability and Mutual Maintenance models lends a nice lens through which the case series' findings may be viewed. Few manuscripts in the clinical literature cite conceptual models - the authors are to be commended for this insightful inclusion. 

Relatedly, Figure 1 does an excellent job illustrating a model of CS syndromes and how each of the 3 cases relate. 

Areas for Improvement. The Introduction section should include a basic description of SGB, which currently appears at the beginning of the Discussion section. It would be challenging for a reader to understand and interpret the authors' findings without an initial such description. 

While the references are relevant, there are significantly more that could be included to further illustrate the authors' familiarity with the topic areas. For instance, the 2020 Rae Olmsted paper (reference number 1) is only one of at least two dozen detailing the use of SGB to treat PTSD. (Also note that, contrary to line 150, the Rae Olmsted publication was with Service members, not Veterans.) Similarly, many more references exist regarding the utility of SGB in treating chronic pain. The authors may also want to note the 1990 publication by Lebovitz, Yarmush, and Lefkowitz, which detailed a case report of an individual suffering from both PTSD and reflex sympathetic dystrophy who was treated with SGB. DOI: 10.1097/00002508-199006000-00015

There are a number of instances where the authors use what I consider to be inflated language. For instance, line 80 states that this case series demonstrates SGB and an effective treatment, however they later (correctly) note that effectiveness cannot be shown in such a study. The authors should attenuate their language here and elsewhere to illustrate their understanding throughout that these data are extremely preliminary. In another instance, lines 195-196 state that the paper's results "provide evidence supporting the use of SGB as a unified treatment approach." This overstates the importance of the paper's findings. 

The sentence spanning lines 62 through 65 is very complicated and would benefit from clarification. Also, references are necessary for the statements regarding cLBP and PTSD both being CSS, and fibromyalgia, myofascial pain syndrome, and CRPS being considered chronic overlapping pain conditions.

It appears the authors relied on CAPS-5 characterization of PTSD symptoms; it would be useful to have a sentence or two describing that assessment. Relatedly, CAPS-5 scores are noted categorically in the text and in Table 1; without understanding how these categorizations were assigned, the symptom severity cannot be interpreted by the reader. (As an aside, it is interesting that the authors reference the CAPS-5, which is clinician-administered, as opposed to the PCL-5, which is a self-report measure and is far more routinely used for research purposes.)

On line 101, the authors introduce the findings for Patient 2, however, as currently described, the data presented are limited such that the case adds nothing to the manuscript. The patient appears to only have data from prior to the procedure, and later at 3 months- but there are no data for the other 3 data points noted for Patients 1 and 3 (namely, 1-day, 1-week, and 6-weeks follow-up). I strongly suggest omitting this patient and focusing on the other 2 patients. 

The authors note on line 164 that the complication rate is approximately 1.7 per 1,000 procedures, a finding reported by Wulf and Maier in 1992 (in German), however that rate was prior to the introduction of fluoroscopic-, and later, ultrasound-guided visualization. This is alluded to in lines 169-171, however it should be noted alongside the 1.7 per 1,000 rate as well so as to underline the fact that fluoro and US have made the procedure safer since then. 

Section 3.3 reads like a "Strengths" section and should probably be re-named as such (given that Section 3.4 is "Limitations"). 

Section 3.5 presents future directions, and the authors correctly note the need for identification of the optimal injectate for SGB. While this varies wildly between providers, the authors may wish to reference Mulvaney, Lynch, and Kotwal's 2015 Clinical Guidelines. 

In sum, this manuscript has numerous opportunities for improvement, however it represents an interesting and potentially compelling way forward for treatment of seemingly disparate-yet-related conditions if revisions are made. 

Author Response

Reviewer 3

This manuscript illustrates findings among 3 individuals with concurrent posttraumatic stress disorder (PTSD) and central sensitization (CS) who were treated with stellate ganglion block (SGB). It presents a rationale for how these conditions relate to one another, and as such, how SGB might be appropriate for both. This is an intriguing subject area for consideration.

Strengths. The description of the Shared Vulnerability and Mutual Maintenance models lends a nice lens through which the case series' findings may be viewed. Few manuscripts in the clinical literature cite conceptual models - the authors are to be commended for this insightful inclusion. 

Relatedly, Figure 1 does an excellent job illustrating a model of CS syndromes and how each of the 3 cases relate. 

Areas for Improvement. The Introduction section should include a basic description of SGB, which currently appears at the beginning of the Discussion section. It would be challenging for a reader to understand and interpret the authors' findings without an initial such description. 

Comment 1: While the references are relevant, there are significantly more that could be included to further illustrate the authors' familiarity with the topic areas. For instance, the 2020 Rae Olmsted paper (reference number 1) is only one of at least two dozen detailing the use of SGB to treat PTSD. (Also note that, contrary to line 150, the Rae Olmsted publication was with Service members, not Veterans.) Similarly, many more references exist regarding the utility of SGB in treating chronic pain. The authors may also want to note the 1990 publication by Lebovitz, Yarmush, and Lefkowitz, which detailed a case report of an individual suffering from both PTSD and reflex sympathetic dystrophy who was treated with SGB. DOI: 10.1097/00002508-199006000-00015

Response 1: Thank you for your comments. We are limited in references due to the case series article type restrictions including on word count, but have added the 1990 publication and updated the Olmsted discussion for service members.

Comment 2: There are a number of instances where the authors use what I consider to be inflated language. For instance, line 80 states that this case series demonstrates SGB and an effective treatment, however they later (correctly) note that effectiveness cannot be shown in such a study. The authors should attenuate their language here and elsewhere to illustrate their understanding throughout that these data are extremely preliminary. In another instance, lines 195-196 state that the paper’s results “provide evidence supporting the use of SGB as a unified treatment approach.” This overstates the importance of the paper’s findings. 

Response 2:  These corrections for inflated language have now been made on page 2 line 80 and page 4 line 197

Comment 3: The sentence spanning lines 62 through 65 is very complicated and would benefit from clarification. Also, references are necessary for the statements regarding cLBP and PTSD both being CSS, and fibromyalgia, myofascial pain syndrome, and CRPS being considered chronic overlapping pain conditions.

Response 3: This has been done.

Comment 4: It appears the authors relied on CAPS-5 characterization of PTSD symptoms; it would be useful to have a sentence or two describing that assessment. Relatedly, CAPS-5 scores are noted categorically in the text and in Table 1; without understanding how these categorizations were assigned, the symptom severity cannot be interpreted by the reader. (As an aside, it is interesting that the authors reference the CAPS-5, which is clinician-administered, as opposed to the PCL-5, which is a self-report measure and is far more routinely used for research purposes.)

Response 4: The CAPS-5 is a structured interview that makes a current diagnosis of PTSD, as well as assesses PTSD symptoms over the past week. Standardized questions are used to target the duration and onset of symptoms, impact of symptoms on daily functioning, and change in symptoms. We have included this as an asterisk under Table 1. Because all authors are clinicians, a clinical assessment tool was used. This was a case series, not a prospective trial.

Assessment of PTSD symptoms was done by the Clinician-Administered PTSD scale for DSM-5 (CAPs-5). The CAPS-5 is a structured interview that makes a current diagnosis of PTSD, as well as assesses PTSD symptoms over the past week. Standardized questions are used to target the duration and onset of symptoms, impact of symptoms on daily functioning, and change in symptoms. The CAPS-5 administration was done over the phone for patient follow ups.

Pain levels were similarly assessed over the phone. Patients were asked to assess their pain from a score of 0-10, with 0 being no pain and 10 being the worst possible pain. The patients were asked to assign a numerical score for each etiology of pain they were were experiencing.

Comment 5: On line 101, the authors introduce the findings for Patient 2, however, as currently described, the data presented are limited such that the case adds nothing to the manuscript. The patient appears to only have data from prior to the procedure, and later at 3 months- but there are no data for the other 3 data points noted for Patients 1 and 3 (namely, 1-day, 1-week, and 6-weeks follow-up). I strongly suggest omitting this patient and focusing on the other 2 patients.   

Response 5:  Table updated

Comment 6: The authors note on line 164 that the complication rate is approximately 1.7 per 1,000 procedures, a finding reported by Wulf and Maier in 1992 (in German), however that rate was prior to the introduction of fluoroscopic-, and later, ultrasound-guided visualization. This is alluded to in lines 169-171, however it should be noted alongside the 1.7 per 1,000 rate as well so as to underline the fact that fluoro and US have made the procedure safer since then. 

Response 6: Added to page 4 line 153-154 and 175-176

Comment 7: Section 3.3 reads like a "Strengths" section and should probably be re-named as such (given that Section 3.4 is "Limitations"). 

Response 7: Thank you, this has been renamed.

Comment 8: Section 3.5 presents future directions, and the authors correctly note the need for identification of the optimal injectate for SGB. While this varies wildly between providers, the authors may wish to reference Mulvaney, Lynch, and Kotwal's 2015 Clinical Guidelines. 

Response 8: Added in Future Directions, reference 22. Thank you.

In sum, this manuscript has numerous opportunities for improvement, however it represents an interesting and potentially compelling way forward for treatment of seemingly disparate-yet-related conditions if revisions are made. 

Round 2

Reviewer 1 Report

Comments and Suggestions for Authors

The authors have addressed my concerns and this will be a nice addition to the field. 

Author Response

Comments 1: The authors have addressed my concerns and this will be a nice addition to the field. 

Response 1: Thank you for your thorough review.  We appreciate your input in enhancing this manuscript.

Reviewer 3 Report

Comments and Suggestions for Authors

The authors are to be commended for making significant improvements to this manuscript. Overall, the language is more appropriate, and several of my initial comments have been addressed. I hope the following notes provide additional clarity for what I believe is a topic that would be of interest to the Clinical Neurology readership.

Comment 1- Line 161 still erroneously reports that the Rae Olmsted study was conducted in Veterans. Please make sure this is corrected anywhere the study is cited, as there is a difference between Veterans and Service members.

Comment 2- Thank you for revising the inflated language. Still, such language appears elsewhere in the manuscript and should be revised similarly. 

Comment 3- This reads much more clearly now and the sentence benefits from the addition of the reference. 

Comment 4- It is still not clear in the text how the CAPS-5 categories were established. What does "severe/markedly elevated" mean? (This would be very simply addressed by noting that the categories of scoring for the CAPS-5 include absent; mild/subthreshold; moderate/threshold; severe/markedly elevated; and extreme/incapacitating. Without inclusion of this information, a reader who is unfamiliar with the CAPS-5 would have no understanding of what "severe/markedly elevated" means.) Additionally, the information about the CAPS-5 should only appear where the CAPS-5 is named. (It is not named in Table 1 so this information does not belong there.) It would be appropriate to put this information between mention of the CAPS-5 in line 83 and the information about the pain scale beginning on line 84. In other words, there is information in lines 84-86 about the pain scale, but no corresponding information about the CAPS-5. 

Comment 5- Thank you for this revision.

Comment 6- Thank you for this revision.

Comment 7- My original suggestion was to rename this section Strengths, like section 3.4 is Limitations. In other words, strengths of the research- just as 3.4 is limitations of the research.

Comment 8- Thank you for this revision.

Lines 193 to 196 are poorly worded, with "based on" and "randomized" used twice each. Importantly, lines 193-194 erroneously state that SGB has been "adopted into clinical practice."

Author Response

Thank you for the additional comments to improve clarity and accessibility to the readership. We have included below only comments that required an additional response.

Comment 1- Line 161 still erroneously reports that the Rae Olmsted study was conducted in Veterans. Please make sure this is corrected anywhere the study is cited, as there is a difference between Veterans and Service members.

Response 1: Thank you for catching this.  This is corrected.

Comment 4- It is still not clear in the text how the CAPS-5 categories were established. What does "severe/markedly elevated" mean? (This would be very simply addressed by noting that the categories of scoring for the CAPS-5 include absent; mild/subthreshold; moderate/threshold; severe/markedly elevated; and extreme/incapacitating. Without inclusion of this information, a reader who is unfamiliar with the CAPS-5 would have no understanding of what "severe/markedly elevated" means.) Additionally, the information about the CAPS-5 should only appear where the CAPS-5 is named. (It is not named in Table 1 so this information does not belong there.) It would be appropriate to put this information between mention of the CAPS-5 in line 83 and the information about the pain scale beginning on line 84. In other words, there is information in lines 84-86 about the pain scale, but no corresponding information about the CAPS-5. 

Response 4: The CAPS-5 is named in the title of Table 1.  However, we agree with the reviewer that it makes more sense to include a description of the tool next to the description of the pain score (lines 84-86) and have done this. 

Comment 7- My original suggestion was to rename this section Strengths, like section 3.4 is Limitations. In other words, strengths of the research- just as 3.4 is limitations of the research.

Response 7: Thank you.  This has been revised.

Comment 8: Lines 193 to 196 are poorly worded, with "based on" and "randomized" used twice each. Importantly, lines 193-194 erroneously state that SGB has been "adopted into clinical practice."

Response 8: Thank you. This duplicate wording has been revised. Though perhaps SGB is not yet widely used or reimbursed by private insurance for PTSD and is not a first-line therapy, we have adopted SGB into clinical practice at our Veterans Affairs hospital and frequently utilize the block for refractory PTSD. In our pain clinic, we accept at least 1 new referral each month to perform SGB for refractory PTSD; it has also been adopted at many other military and private practice locations throughout the United States. As noted, there is national guidance provided by the VA for SGBs in practice throughout this large healthcare system, so we feel strongly that this is an accurate statement. We have included a link to the PDF document again, here, for reference to SGB as a potential treatment for refractory cases of PTSD: Stellate Ganglion Block for PTSD at VA. We understand that this change is new, and perhaps not all providers are informed that this is now an available therapy for PTSD, which is why we seek to inform others of these developments.

Thank you again for taking the time to review our manuscript and enhance its quality.